# Trends in low-density lipoprotein cholesterol goal achievement and changes in lipid-lowering therapy after incident atherosclerotic cardiovascular disease: Danish cohort study

**Annette Kjær Ersbøll**[1]*, **Marie Skov Kristensen**[1], **Mads Nybo**[2], **Simone Møller Hede**[3], **Kristian Handberg Mikkelsen**[4], **Gunnar Gislason**[1,5,6,7], **Mogens Lytken Larsen**[8], **Anders Green**[3,9]

1 National Institute of Public Health, University of Southern Denmark, Copenhagen, Denmark, 2 Department of Clinical Biochemistry, Odense University Hospital, Odense, Denmark, 3 Institute of Applied Economics and Health Research, Copenhagen, Denmark, 4 Sanofi-Aventis Denmark A/S, Copenhagen, Denmark, 5 Department of Cardiology, The Cardiovascular Research Centre, Copenhagen University Hospital Herlev and Gentofte, Gentofte, Denmark, 6 Faculty of Health and Medical Sciences, Copenhagen University, Copenhagen, Denmark, 7 The Danish Heart Foundation, Copenhagen, Denmark, 8 Department of Cardiology, Aalborg University, Aalborg, Denmark, 9 Steno Diabetes Center Odense, Odense University Hospital and University of Southern Denmark, Odense, Denmark

* ake@sdu.dk

**Data Availability Statement:** Data cannot be shared because of data privacy regulations by

## Abstract

### Background

We aimed to investigate trends in low-density lipoprotein cholesterol (LDL-C) goal achievement (LDL-C<1.8 mmol/L, equivalent to 70 mg/dL), initiation of lipid-lowering therapy (LLT) and changes in LLT intensity in individuals with atherosclerotic cardiovascular disease (ASCVD) at very high risk of recurrent cardiovascular disease.

### Methods

A cohort study design was used including individuals with incident ASCVD and LDL-C$\geq$1.8 mmol/L in 2010–2015. Data were obtained from national, population-based registers (patient, prescription, income, and laboratory).

### Results

We included 11,997 individuals. Acute myocardial infarction, ischemic stroke and stable angina pectoris accounted for 79.6% of the qualifying ASCVD events. At inclusion, 37.2% were in LLT. Mean LDL-C before or during ASCVD hospitalization was 3.1 mmol/L (120 mg/dL). LDL-C goal achievement increased within the first two years after inclusion from 40.5% to 50.6%. LLT initiation within the first 90 days increased from 48.6% to 56.0%. Initiation of intensive LLT increased from 9.6% to 32.8%. The largest change in LLT intensity was seen in the period 180 days before to 90 days after discharge with 2.2% in 2010 to 12.1% in 2015.

Statistics Denmark. The datasets generated and analyzed during the current study are not available. The researchers have access to data at servers at Statistics Denmark. According to the regulations at Statistics Denmark, it is not allowed to extract data from the servers at Statistics Denmark. Therefore, data are not available. The Act on Processing of Personal Data is describing the regulations regarding use of individual-level data for research in Denmark. According to this Act, individual level data from Statistics Denmark are not delivered to any external firm, institution, or person. Instead, datasets and linkages between datasets are stored at Statistics Denmark. Researchers employed at specific authorised environments can establish remote online access to these datasets stored at Statistics Denmark. The researcher gets online access to the datasets. Although the researchers may get access to rather detailed individual level data, they are only allowed to publish statistical analyses and results at an aggregate level where no single person or enterprise may be identified. For security reasons, only researchers employed at authorised research institutions can get access to individual level data at Statistics Denmark. And only permanent research institutions with a responsible leader and several researchers can be authorised.

**Funding:** This work was supported by Sanofi Aventis Denmark A/S (www.sanofi.dk) and Applied Economics and Health Research (ApHER) (appliedeconomics.dk). The paper is based on data originating from a study conducted for Applied Economics and Health Research (ApHER) as an independent research institute and funded by Sanofi Aventis Denmark A/S. The sponsor (Sanofi) was involved in the conceptualization and the final review and editing of the manuscript.

**Competing interests:** I have read the journal's policy and the authors of this manuscript have the following competing interests: When this study was performed, KH Mikkelsen was an employee of Sanofi Aventis Denmark A/S A Green and SM Hede were associated with ApHER that received funding from Sanofi Aventis Denmark A/S during the conduct of the study, and they declare no other relationships or activities outside the submitted work GG declare that he has received funding from Bristol Myers Squibb, Bayer, Boehringer Ingelheim and Pfizer outside the submitted work. Page 2 All other authors declare no conflict of interest. This does not alter our adherence to PLOS ONE policies on sharing data and materials.

## Conclusion

LDL-C goal achievement within the first 2 years after inclusion increased from 40.5% in 2010 to 50.6% in 2015. LLT initiation within the first year after inclusion increased, especially for intensive LLT, although only one third initiated intensive LLT in 2015. Despite trends show improvements in LDL-C goal achievement, 49.4% of individuals at very high risk of a CV event did not achieve the LDL-C goal within 2 years after ASCVD hospitalization.

## Introduction

Atherosclerotic cardiovascular disease (ASCVD) is a leading cause of death with ischemic heart disease and stroke accounting for 27% of deaths worldwide [1–3]. Furthermore, previous ASCVD increases the risk of recurrent cardiovascular (CV) events [4, 5].

Several lines of evidence together strongly suggest that low-density lipoprotein cholesterol (LDL-C) is causal in the development of ASCVD [6]. Therefore, the European Society of Cardiology (ESC)/European Atherosclerosis Society (EAS) guidelines recommended LDL-C goals<1.8 mmol/L (equivalent to 70 mg/dL) in individuals with established CV disease [7]. The 2019 revised guidelines recommend an even lower treatment goal as LDL-C<1.4 mmol/L (55 mg/dL) for individuals at very high risk or with ASCVD [8].

Effective lipid-lowering therapy (LLT) is widely available with statins being the cornerstone. Treatment efficacy is well established [9], but the statin dose is important to effectively reduce the LDL-C level and thereby the risk of ASCVD and CV events. New drugs for managing dyslipidemia have recently emerged, such as proprotein convertase subtilisin/kexin type 9 (PCSK9) inhibitors [10, 11].

A considerable proportion of individuals with ASCVD do not achieve LDL-C<1.8 mmol/L (70 mg/dL) [12–19]. Furthermore, many individuals with an elevated LDL-C do not receive LLT doses according to guidelines [18, 20–22].

LDL-C goal achievement and LLT intensity in high-risk CV populations have improved over the last decades [23–27]. However, these studies were either performed using cross-sectional data, other high-risk populations than ASCVD, small sample sizes, limited recent data or short times series.

The aim of the study was to investigate LDL-C goal achievement and changes in LLT intensity in a population with an incident ASCVD diagnosed in 2010 to 2015 and LDL-C≥1.8 mmol/L (70 mg/dL) before or during ASCVD hospitalization.

Specifically, we examined 1) if the proportion of individuals achieving LDL-C<1.8 mmol/L (70 mg/dL) within the first two years after diagnosis increased during the study period, 2) if the proportion of individuals initiating LLT or initiating intensive LLT before or within the first year after diagnosis increased during the study period, and 3) if the proportion of individuals initiating LLT or changing towards more intensive LLT within the first year after diagnosis increased during the study period.

## Materials and method

### Study design and population

This population-based cohort study included individuals with a first-ever incident diagnosis of ASCVD in the study period 2010–2015. Index date was defined as date of hospitalization with a qualifying ASCVD diagnosis. ASCVD included acute myocardial infarction (AMI), ischemic

stroke (IS), unstable angina pectoris (UA) together with a procedure of coronary angiography (CAG), stable angina pectoris (SA) together with a procedure of CAG or computerized tomography coronary angiography (CT-CAG), and peripheral artery disease (PAD). Furthermore, a procedure of coronary artery bypass grafting (CABG), and a procedure of percutaneous coronary intervention (PCI) qualified for ASCVD regardless of discharge diagnoses connected with the procedure. We used primary diagnoses or primary procedures recorded at a hospital except for AMI and stroke where also secondary diagnoses were included (S1 Table in S1 File for ICD-8/10 and procedure codes). In case of two or more ASCVD events at the same date, we coded ASCVD in the following order: AMI, IS, UA, SA, PAD, CABG, and PCI.

### Inclusion and exclusion criteria

To be included in the study, individuals should be residents of Funen Island, Denmark, at index date. The population in the Region of Southern Denmark including Funen is a representative sample of the entire Danish population [28] regarding demographic and socioeconomic characteristics, healthcare utilization, and use of medication. Furthermore, individuals should have at least one LDL-C measurement up to 18 months before index date or during ASCVD hospitalization (i.e., index date to date of discharge).

   We excluded individuals with an ASCVD diagnosis in the period 1977–2009 to ascertain truly first-ever ASCVD cases. Individuals were also excluded if the unique personal identification number was invalid, if the address was outside the Funen municipality at date of the ASCVD diagnosis, or if they died during the ASCVD hospitalization. Finally, individuals were excluded if the LDL-C measurement during ASCVD hospitalization was <1.8 mmol/L (70 mg/dL) or no LDL-C measurements were available during or before ASCVD hospitalization.

### Data sources

We used national, population-based registers including the Danish National Patient Register [29], the Danish National Prescription Register [30], the Danish income registers [31] and the Laboratory databases at Odense University Hospital [32]. Individual-level linkage of data from the registers and the database was possible due to the unique personal identification number assigned to all individuals at birth or immigration and registered in the Danish Civil Registration System [33].

### Low-density lipoprotein cholesterol (LDL-C)

The collection of blood samples, laboratory analyses and database storage have been described previously [20]. Briefly, blood samples collected by the general practitioner or at hospital wards were analyzed at hospital-based laboratories at Funen, Denmark. The test results were stored in a laboratory information system (BCC) administrated by Odense University Hospital. The lipid measurements included LDL-C, high-density lipoprotein cholesterol (HDL-C), total cholesterol and triglycerides. LDL-C is given as mmol/L and can be converted to mg/dL by multiplying with 38.6. LDL-C goal achievement was defined as LDL-C<1.8 mmol/L in periods after hospital admission (1–180 days, 181–356 days, 366–730 days).

### Lipid-lowering therapy (LLT)

Type and intensity of LLT were defined based on prescription redemptions (S1 Table in S1 File for ATC codes) as described previously [20]. Briefly, type of LLT was defined as statins, ezetimibe, other non-statins, and combination therapies. Intensity of LLT was categorized as: no LLT, moderate LLT and intensive LLT [5, 7, 26, 34]. We defined intensive LLT by having 1)

a minimum of two prescription redemptions of one combinational drug; 2) a minimum of two prescription redemptions of statins (80 mg Simvastatin, 40–80 mg Atorvastatin or 20–40 mg Rosuvastatin); 3) a minimum of two prescription redemptions of statins (all doses and types) and one ezetimibe prescription redemption; or 4) a minimum of two prescription redemptions of statins (all doses and types) and one other non-statin prescription redemption (i.e. not ezetimibe). We defined moderate LLT intensity as having LLT prescription redemption not included in the definition of intensive LLT. Initiation of LLT therapy was defined as having a prescription redemption of the above-described drugs before hospital admission (180 days before admission) and in periods after admission (admission-90 days after discharge, 91–180 days after discharge, 181–365 days after discharge). Initiation of LLT therapy later than 90 days after discharge was included to capture potentially late initiations, although clinical guidelines recommend initiation of individuals at very high risk or with ASCVD.

## Follow-up and censoring

Individuals were followed for a maximum of two years with censoring at the end of the study period (2 years after discharge), at death, or if they moved out of Funen, whichever came first.

## Baseline characteristics

Sociodemographic characteristics included were age (30–39, 40–49, 50–59, 60–69, 70–79, ≥80), gender, cohabitation (yes, no), ethnicity (ethnic Danes, emigrants or descendants from Western or Non-western countries), and income (quartiles). The clinical variables included were ASCVD event at index date (AMI, IS, UA+CAG, SA+CAG/CT-CAG, PAD, CABG, PCI), comorbidity (diabetes and chronic kidney disease), LLT (statins, ezetimibe, other non-statins, combination therapy, no treatment), LLT intensity (intensive, moderate, no) and lipid measurements (LDL-C, HDL-C, total cholesterol, triglycerides) (S1 Table in S1 File for ICD-8, ICD-10, procedure, and ATC codes).

## Statistical analysis

We used basic descriptive statistics to present baseline characteristics of the study population, the development in LDL-C goal achievement, development in initiation of LLT therapy, and changes in LLT therapy during the study period.

We used a logistic regression model to test if the proportion of individuals achieving LDL-C goal increased during the study period. Goal achievement was a binary outcome. Similarly, we used a logistic regression model to test if a larger proportion of individuals initiated (or continued) LLT therapy after index date. LLT prescription redemption was a binary outcome. In both analyses, year of hospitalization due to the qualifying ASCVD event was included as a continuous variable to test for trend. Linearity between year of ASCVD event and log(odds) of each of the binary outcomes was evaluated visually by the parameters estimates of year as a categorical variable versus calendar year.

All analyses were performed with SAS 9.4. A 5% significance level was applied.

## Supplementary analyses

In the first supplementary analysis, the treatment goal for LDL-C for individuals at very high risk or with ASCVD was reduced to 1.4 mmol/L (55 mg/mL) as recommended in the 2019 revised guidelines [8]. The descriptive analysis of development of LDL-C goal achievement in the study period was repeated with LDL-C treatment goal at 1.4 mmol/L.

The second supplementary analysis examined sociodemographic and clinical characteristics associated with initiation of LLT therapy (moderate or intensive) between admission and 90 days after discharge among individuals with no LLT therapy before admission using a logistic regression model. LLT initiation was a binary outcome. In this analysis, lipid measurements (LDL-C, HDL-C, total cholesterol (TC) and triglycerides) were included as categorical variables in three categories (lowest 25%, middle 50%, and highest 25%).

A third supplementary analysis of trends in initiation of LLT therapy was performed stratified by type of qualifying event of ASCVD (AMI and IS versus remaining diagnoses and/or procedures). This was performed to evaluate if initiation of LLT therapy varied depending on the severity of the ASCVD event.

## Results

We identified a total of 39,608 individuals with a first-ever diagnosis of ASCVD in the period 1.1.1977–31.12.2015 among residents of Funen Island, Denmark. We excluded 22,647 individuals with ASCVD before the study period (before 1.1.2010), resulting in a total of 16,961 individuals with incident ASCVD in the study period 2010–2015. Among individuals who survived to discharge (15,396), 10% lack LDL-C measurements before and/or during hospitalization (S2 Table in S1 File). After exclusions for various reasons (Fig 1 and S2 Table in S1 File), the final study population comprised 11,997 individuals who were discharged alive and with an LDL-C≥1.8 mmol/L (70 mg/dL) before or during ASCVD hospitalization.

### Baseline characteristics

The study population was characterized by a mean age of 68.2 years, 59.7% males, 56.0% cohabitants, and 94.5% ethnic Danes (Table 1).

AMI, IS, and SA-CAG/CT-CAG accounted together for 79.6% of the qualifying ASCVD events. A history of diabetes and chronic kidney disease at index date was observed among 14.8% and 3.7% of the study population, respectively. In total, 4464 individuals (37.2%) had redeemed at least one LLT prescription up to 180 days before ASCVD hospitalization, of which 350 (7.8%) had redeemed prescriptions of intensive LLT. Mean LDL-C during or before ASCVD hospitalization was 3.1 mmol/L (120 mg/dL). The mean HDL-C, total cholesterol and triglycerides were 1.4 mmol/L, 5.2 mmol/L and 1.8 mmol/L, respectively.

Stratified by time of LDL-C measurement (before or during ASCVD hospitalization) only minor differences were seen, except for the distribution of the qualifying ASCVD event. Among individuals with AMI and IS being the qualifying ASCVD event, 80.5% and 72.6% had an LDL-C measurement during ASCVD hospitalization, while only 17.2% and 19.1% of the individuals with SA+CAG/CT-CAG and PAD as the qualifying events had an LDL-C measurement during ASCVD hospitalization.

### Development in LDL-C goal achievement (LDL-C<1.8 mmol/L)

The proportion of individuals who achieved LDL-C<1.8 mmol/L (70 mg/dL) during the first 730 days after ASCVD hospitalization increased from 40.5% in 2010 to 50.6% in 2015 (p<0.001) (Fig 2 and S3 Table in S1 File). Stratified by days after ASCVD hospitalization, the proportion of individuals who achieved LDL-C<1.8 mmol/L (70 mg/dL) within 180 days after ASCVD hospitalization increased from 13.1% in 2010 to 23.3% in 2015 (p<0.001). A small increase was also seen in the period 181–365 days after ASCVD hospitalization (5.9% in 2010 to 9.0% in 2015). No significant difference was seen in the period 366–735 days after ASCVD hospitalization.

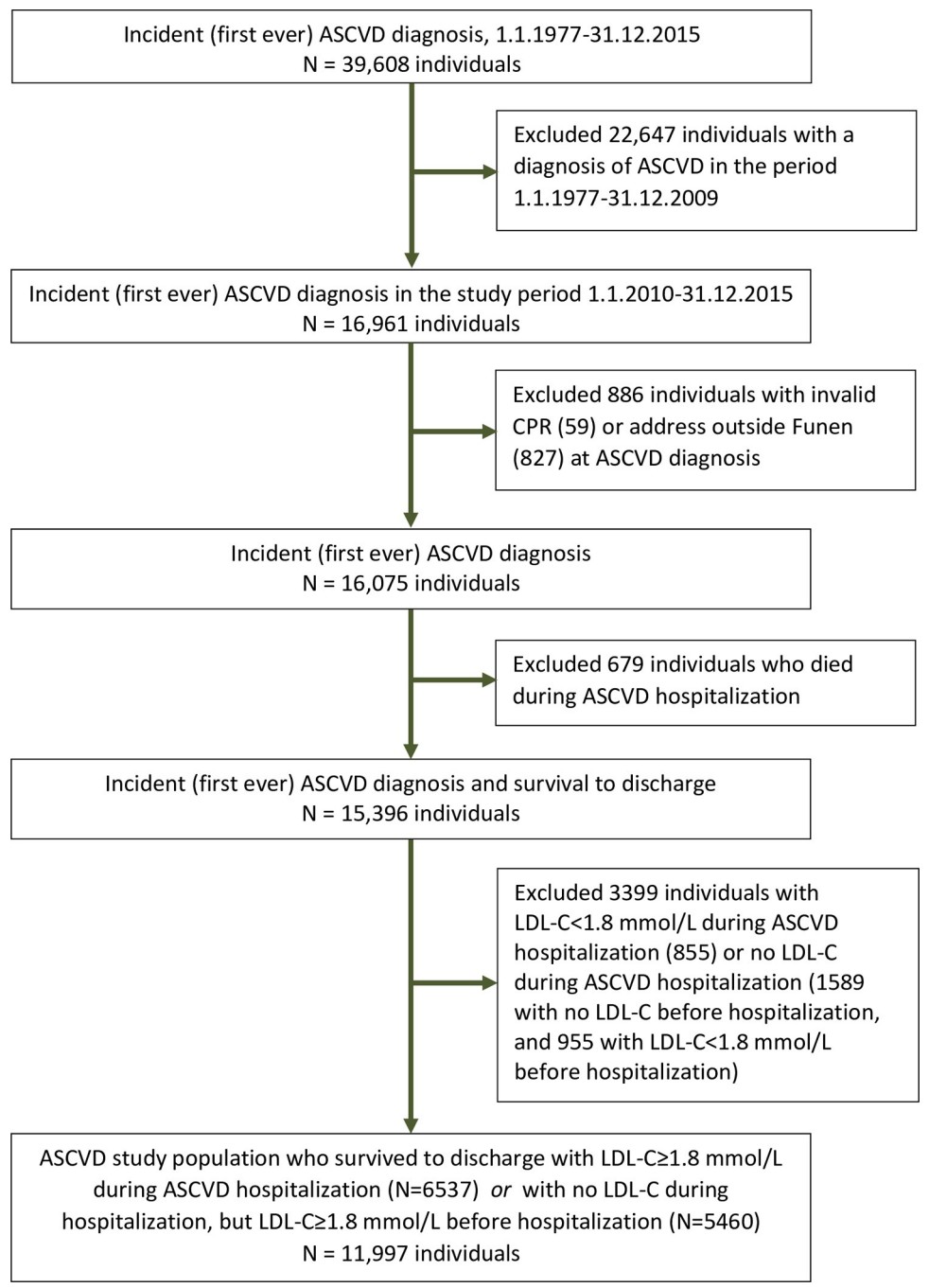

**Fig 1. Flow diagram illustrating the construction of the population of individuals with incident atherosclerotic cardiovascular disease (ASCVD) and low-density lipoprotein cholesterol (LDL-C) ≥1.8 mmol/L (equivalent to 70 mg/dL) (January 1, 2010 to December 31, 2015) based on data from nation-wide population registers and a laboratory database.** CPR, unique personal identification number.

## Development in initiating LLT and initiating intensive LLT

The proportion of individuals in LLT (i.e., at least one LLT prescription redemption) before index date did not change during the study period (19.2–22.3% in LLT, p = 0.36) (Fig 3A and S4 Table in S1 File). The proportion of individuals who initiated LLT within the first 90 days

**Table 1. Sociodemographic and clinical baseline characteristics of individuals (N = 11,997) at date of the incident atherosclerotic cardiovascular disease (ASCVD population) stratified by low-density lipoprotein cholesterol (LDL-C) ≥1.8 mmol/L (70 mg/dL) either during or before ASCVD hospitalization given by number and percentage (N, %) if nothing else is indicated.** Percentages are given by rows, except for Total where percentages are given within the column.

| | LDL-C ≥ 1.8 mmol/L (equivalent to 70 mg/dL) | | |
|---|---|---|---|
| | During or before ASCVD (index) hospitalization | | |
| | **Total** | **During** | **Before** |
| | N = 11,997 | 6537 (54.5%) | 5460 (45.5%) |
| Qualifying ASCVD event at index date | | | |
| AMI | 2949 (24.6%) | 2375 (80.5%) | 574 (19.5%) |
| IS | 4159 (34.7%) | 3019 (72.6%) | 1140 (27.4%) |
| UA+CAG | 330 (2.8%) | 171 (51.8%) | 159 (48.2%) |
| SA+CAG/CT-CAG | 2431 (20.3%) | 418 (17.2%) | 2013 (82.8%) |
| PAD | 1524 (12.7%) | 291 (19.1%) | 1233 (80.9%) |
| CABG | 221 (1.8%) | 114 (51.6%) | 107 (48.4%) |
| PCI | 383 (3.2%) | 149 (38.9%) | 234 (61.1%) |
| Calendar year of qualifying ASCVD event | | | |
| 2010 | 2114 (17.6%) | 1064 (50.3%) | 1050 (49.7%) |
| 2011 | 2142 (17.9%) | 1107 (51.7%) | 1035 (48.3%) |
| 2012 | 2099 (17.5%) | 1033 (49.2%) | 1066 (50.8%) |
| 2013 | 1935 (16.1%) | 982 (50.8%) | 953 (49.2%) |
| 2014 | 1840 (15.3%) | 1141 (62.0%) | 699 (38.0%) |
| 2015 | 1867 (15.6%) | 1210 (64.8%) | 657 (35.2%) |
| Age (years), mean (SD) | 68.2 (13.0) | 67.5 (13.7) | 70.0 (12.1) |
| Age group (years) | | | |
| <40 | 187 (1.6%) | 143 (76.5%) | 44 (23.5%) |
| 40–49 | 833 (6.9%) | 545 (65.4%) | 288 (34.6%) |
| 50–59 | 1968 (16.4%) | 1131 (57.5%) | 837 (42.5%) |
| 60–69 | 3241 (27.0%) | 1694 (52.3%) | 1547 (47.7%) |
| 70–79 | 3315 (27.6%) | 1658 (50.0%) | 1657 (50.0%) |
| ≥80 | 2453 (20.5%) | 1366 (55.7%) | 1087 (44.3%) |
| Gender | | | |
| Male | 7157 (59.7%) | 3970 (55.5%) | 3187 (44.5%) |
| Female | 4840 (40.3%) | 2567 (53.0%) | 2273 (47.0%) |
| Cohabitation | | | |
| Yes | 6715 (56.0%) | 3626 (54.0%) | 3089 (46.0%) |
| No | 5282 (44.0%) | 2911 (55.1%) | 2371 (44.9%) |
| Ethnicity | | | |
| Denmark | 11,333 (94.5%) | 6175 (54.5%) | 5158 (45.5%) |
| Western | 378 (3.2%) | 204 (54.0%) | 174 (46.0%) |
| Non-western | 286 (2.4%) | 158 (55.2%) | 128 (44.8%) |
| Comorbidity | | | |
| Diabetes mellitus | 1775 (14.8%) | 809 (45.6%) | 966 (54.4%) |
| Chronic kidney disease | 439 (3.7%) | 193 (44.0%) | 246 (56.0%) |
| Socioeconomic position (quartiles based on income) [a] | | | |
| Q$_1$ (lowest 25%) | 2973 (24.8%) | 1614 (54.3%) | 1359 (45.7%) |
| Q$_2$ | 2970 (24.8%) | 1615 (54.4%) | 1355 (45.6%) |
| Q$_3$ | 2949 (24.6%) | 1573 (53.3%) | 1376 (46.7%) |
| Q$_4$ (highest 25%) | 3105 (25.9%) | 1735 (55.9%) | 1370 (44.1%) |
| Lipid lowering therapy within 180 days before ASCVD hospitalization [b] | | | |
| Statins | 4313 (36.0%) | 1534 (35.6%) | 2779 (64.4%) |

(*Continued*)

**Table 1.** (Continued)

| | LDL-C $\geq$ 1.8 mmol/L (equivalent to 70 mg/dL) | | |
| --- | --- | --- | --- |
| | During or before ASCVD (index) hospitalization | | |
| | **Total** | **During** | **Before** |
| Ezetimibe | 74 (0.6%) | 30 (40.5%) | 44 (59.5%) |
| Other non-statins | 49 (0.4%) | 21 (42.9%) | 28 (57.1%) |
| Combination therapies | 28 (0.2%) | 11 (39.3%) | 17 (60.7%) |
| No treatment | 7533 (62.8%) | 4941 (65.6%) | 2592 (34.4%) |
| Intensity of lipid lowering therapy within 180 days before ASCVD hospitalization | | | |
| No LLT | 7533 (62.8%) | 4941 (65.6%) | 2592 (34.4%) |
| Moderate LLT[c] | 4114 (34.3%) | 1467 (35.7%) | 2647 (64.3%) |
| Intensive LLT[d] | 350 (2.9%) | 129 (36.9%) | 221 (63.1%) |
| Lipid measurements within 18 months before ASCVD hospitalization[e], N (%), mean (SD) | | | |
| LDL-C | 9721 (81.0%), | 4261 (43.8%), | 5460 (56.2%), |
| | 3.09 (0.96) | 3.19 (1.01) | 3.01 (0.91) |
| HDL-C | 9719 (81.0%), | 4262 (43.9%), | 5457 (56.1%), |
| | 1.36 (0.42) | 1.36 (0.42) | 1.36 (0.43) |
| Total cholesterol | 9766 (81.4%), | 4306 (44.1%), | 5460 (55.9%), |
| | 5.21 (1.14) | 5.31 (1.17) | 5.13 (1.12) |
| Triglycerides | 9723 (81.0%), | 4263 (43.8%), | 5460 (56.2%), |
| | 1.75 (1.16) | 1.76 (1.14) | 1.75 (1.18) |

ASCVD: Atherosclerotic cardiovascular disease; AMI: Acute Myocardial Infarction; UA: Unstable angina pectoris; CAG: Coronary angiography; SA: Stable angina pectoris; PCI: Percutaneous coronary intervention; CABG: Coronary artery bypass grafting; PAD: Peripheral artery disease; IS: Ischemic stroke.

[a] Socioeconomic position is derived as quartiles based on income. Quartiles are estimated for combinations of sex (male, female) and age (age<65 years, age≥65 years), for each year, separately.

[b] The latest prescription of LLT redeemed is identified up to 180 days prior index date.

[c] Persons who are treated with LLT, but who are not eligible for the group of "intensive treatment" or "no LLT" treatment.

[d] Intensive LLT including statins with ezetimibe. To identify LLT intensity, prescription redemptions of LLT are identified up to 180 days prior index date.

[e] The latest available lipid measurement is identified up to 18 months prior to index date.

after index date increased from 48.6% in 2010 to 56.0% in 2015 (p<0.001). No differences were seen in the proportion of individuals who initiated LLT 91–180 days (2.2% to 4.4% initiated LLT, p = 0.063) or 181–365 days after index date (2.0% to 3.1% initiated LLT, p = 0.78). Overall, 33.3% to 39.4% did not initiate LLT within 365 days after index date.

The proportion of individuals in intensive LLT before index date increased from 2.4% in 2010 to 3.6% in 2015 (p<0.001) (Fig 3B and S4 Table in S1 File). The proportion of individuals who initiated intensive LLT within the first 90 days after index date increased from 2.2% in 2010 to 12.6% in 2015 (p<0.001). Within 91–180 days and 181–365 days after index date, the proportion of individuals who initiated intensive LLT increased from 0.9% in 2010 to 3.6% in 2015 (p<0.001) and from 4.3% in 2010 to 17.3% in 2015 (p<0.001).

### Development in changing treatment towards more intensive LLT

The largest change in LLT intensity was seen among individuals who changed from none or moderate LLT to intensive LLT (Fig 4A–4C and S5 Table in S1 File). In the period 180 days before index date to 90 days after discharge, the proportion of individuals who changed to intensive LLT increased from 2.2% among individuals with index date in 2010 to 12.1% among individuals with index date in 2015 (p<0.001). In the period 181–365 days after discharge, the proportion of individuals who changed to intensive LLT increased from 6.7%

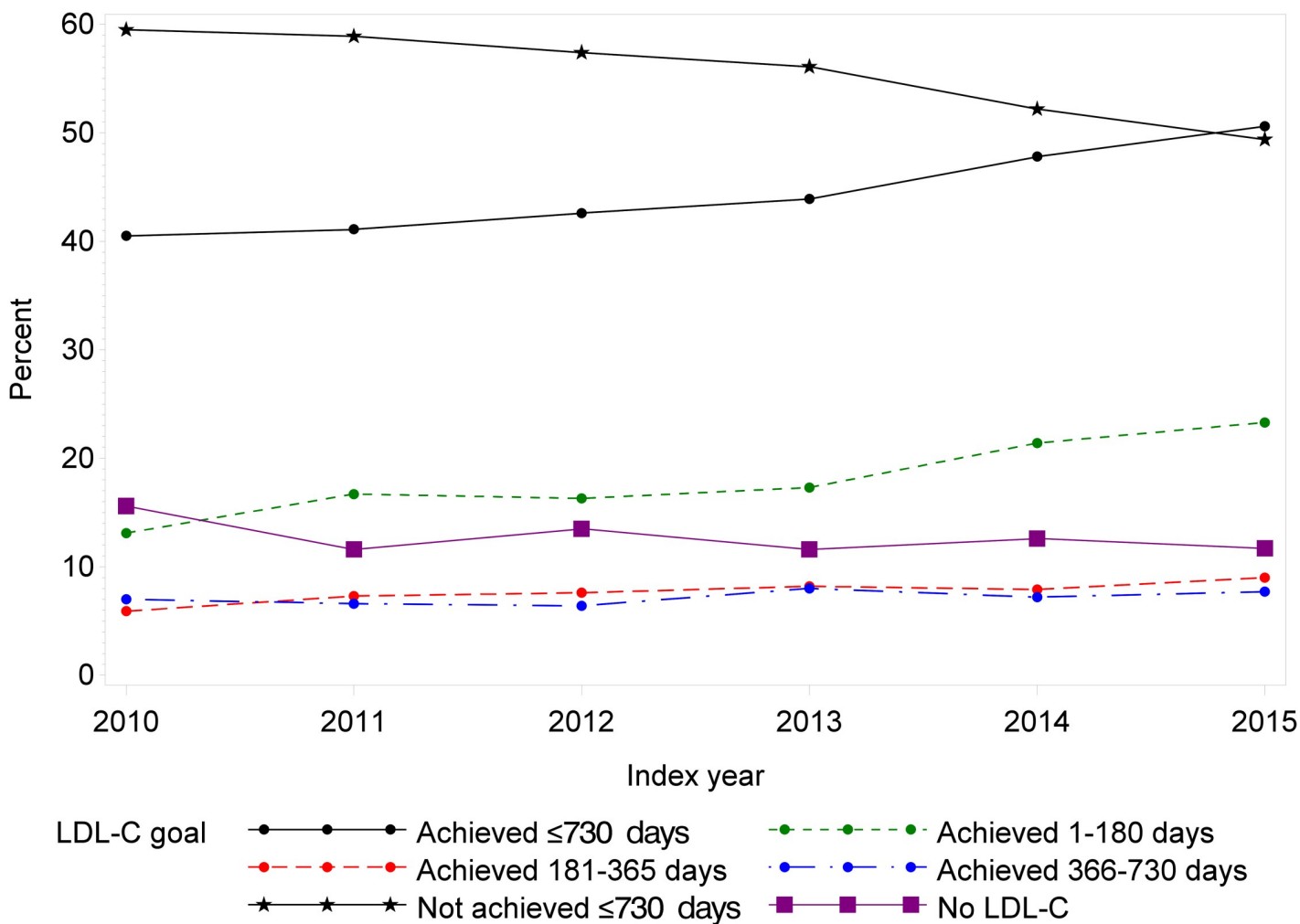

**Fig 2. Development of LDL-C goal achievement in the ASCVD population during the study period (1 January 2010–31 December 2015) given as proportion of individuals who reach treatment goals (LDL-C<1.8 mmol/L, equivalent to 70 mg/dL) within 1–180, 181–365 and 366–730 days after discharge, N = 15,193 individuals.**

among individuals with index date in 2010 to 24.4% among individuals with index date in 2015 (p<0.001). The proportion of individuals with no changes in LLT intensity decreased during the study period in all three periods. In the period 180 days before index date to 90 days after discharge, the proportion of individuals with no changes in LLT intensity decreased from 49.0% among individuals with index date in 2010 to 42.0% among individuals with index date in 2015 (p<0.001). In the period 181–365 days after discharge, the proportion of individuals with no changes in LLT intensity decreased significantly from 71.8% among individuals with index date in 2010 to 58.0% among individuals with index date in 2015 (p<0.001).

## Supplementary analyses

In the first supplementary analysis, the treatment goal for LDL-C for individuals at very high risk or with ASCVD was reduced to 1.4 mmol/L (55 mg/mL) as recommended in the 2019 revised guidelines [8]. Only 23.8–29.3% of the individuals achieved the goal within 730 days. It is further seen that LDL-C goal achievement within the first 180 days increased from 4.9% in 2010 to 9.4% in 2015 (S6 Table in S1 File).

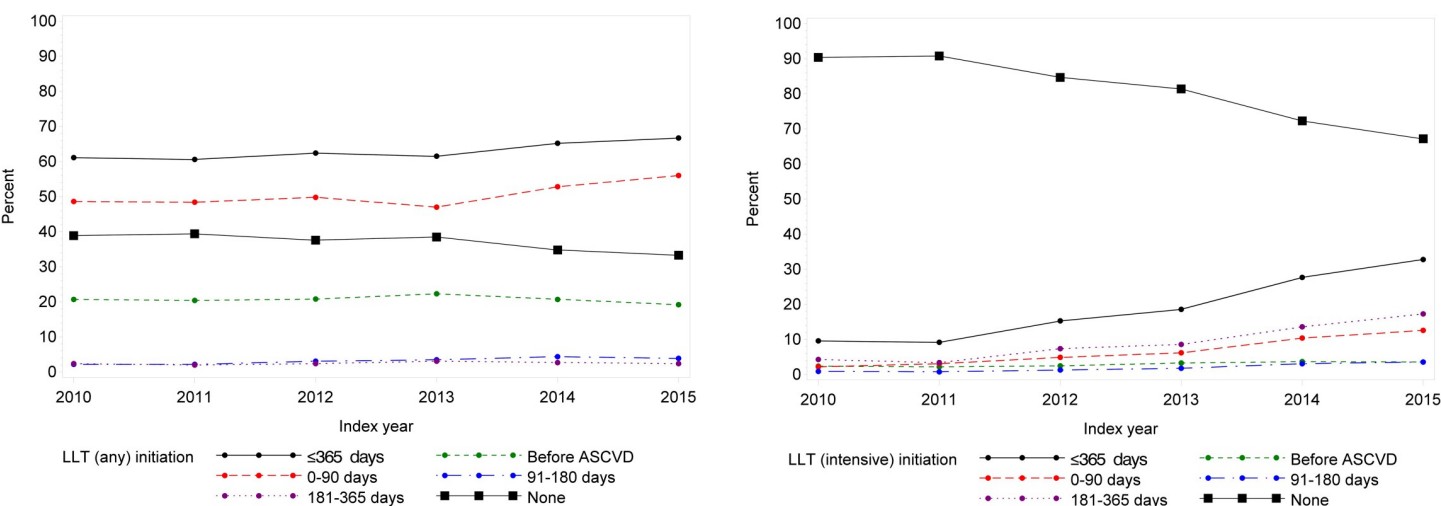

**Fig 3.** Development of LLT treatment patterns in the ASCVD population during the study period (1 January 2010–31 December 2015) given as proportion of individuals who initiated treatment with (A) LLT and (B) intensive LLT before admission, from admission to 90 days after discharge, during 91–180 days after discharge and during 181–365 days after discharge, N = 11,997 individuals with an incident ASCVD.

In the second supplementary analysis, sociodemographic and clinical characteristics associated with initiation of LLT therapy (moderate or intensive) among individuals with no LLT therapy before admission was examined. Odds for initiation of LLT was highest for individuals with AMI or IS as the qualifying ASCVD event at index date (AMI: OR = 10.84 (95% CI: 9.41; 12.48); IS: OR = 8.77 (95% CI: 7.65; 10.05)), younger age (<40 years: OR = 3.95 (95% CI: 2.48; 6.28); 40–49 years: OR = 2.85 (95% CI: 2.38; 3.41); 50–59 years: OR = 1.51 (95% CI: 1.34; 1.71)), without comorbidity (not diabetes: OR = 3.12 (95% CI: 2.75; 3.53); not CKD: OR = 2.29 (95% CI: 1.78; 2.94)) and no LDL-C measurements before admission (OR = 13.17 (95% CI: 11.18; 15.51)) (S7 Table in S1 File). Odds for initiation of LLT was high for individuals with a moderate to high LDL-C measurement (LDL-C at 2.4–3.8: OR = 6.99 (95% CI: 6.13; 7.96) and LDL-C>3.8: OR = 18.38 (95% CI: 15.73; 21.46)), and with a moderate to high total cholesterol, TC (TC at 4.4–59: OR = 4.14 (95% CI: 3.69; 4.65) and TC>5.9: OR = 9.75 (95% CI: 8.50; 11.18)).

The third supplementary analysis examined trends in initiation of LLT therapy stratified by type of ASCVD event (AMI and IS versus remaining diagnoses and/or procedures). A remarkably higher proportion of individuals with AMI or IS initiated LLT therapy between admission and 90 days after discharge compared with the remaining ASCVD diagnoses (S8 Table in S1 File). Among individuals with AMI or IS, the proportion initiating LLT between admission and 90 days after discharge increased during the study period from 67.4% to 74.6%. Among individuals with other ASCVD diagnoses or procedures (SA, UA, PAD, CABG, PCI), the proportion initiating LLT between admission and 90 days after discharge also increased during the study period, however, from 23.1% to 25.0%.

## Discussion

The main findings of the study are an overall improvement in LDL-C goal achievement and an increased LLT initiation for individuals with index date in 2010–2015.

LDL-C goal achievement (i.e., LDL-C<1.8 mmol/L (70 mg/dL)) within the first two years after ASCVD hospitalization increased and the largest increase was seen 1–180 days after ASCVD discharge. However, despite an improvement in LDL-C goal achievement, a large proportion of individuals did not reach the goal within 730 days after index date. In relation to the

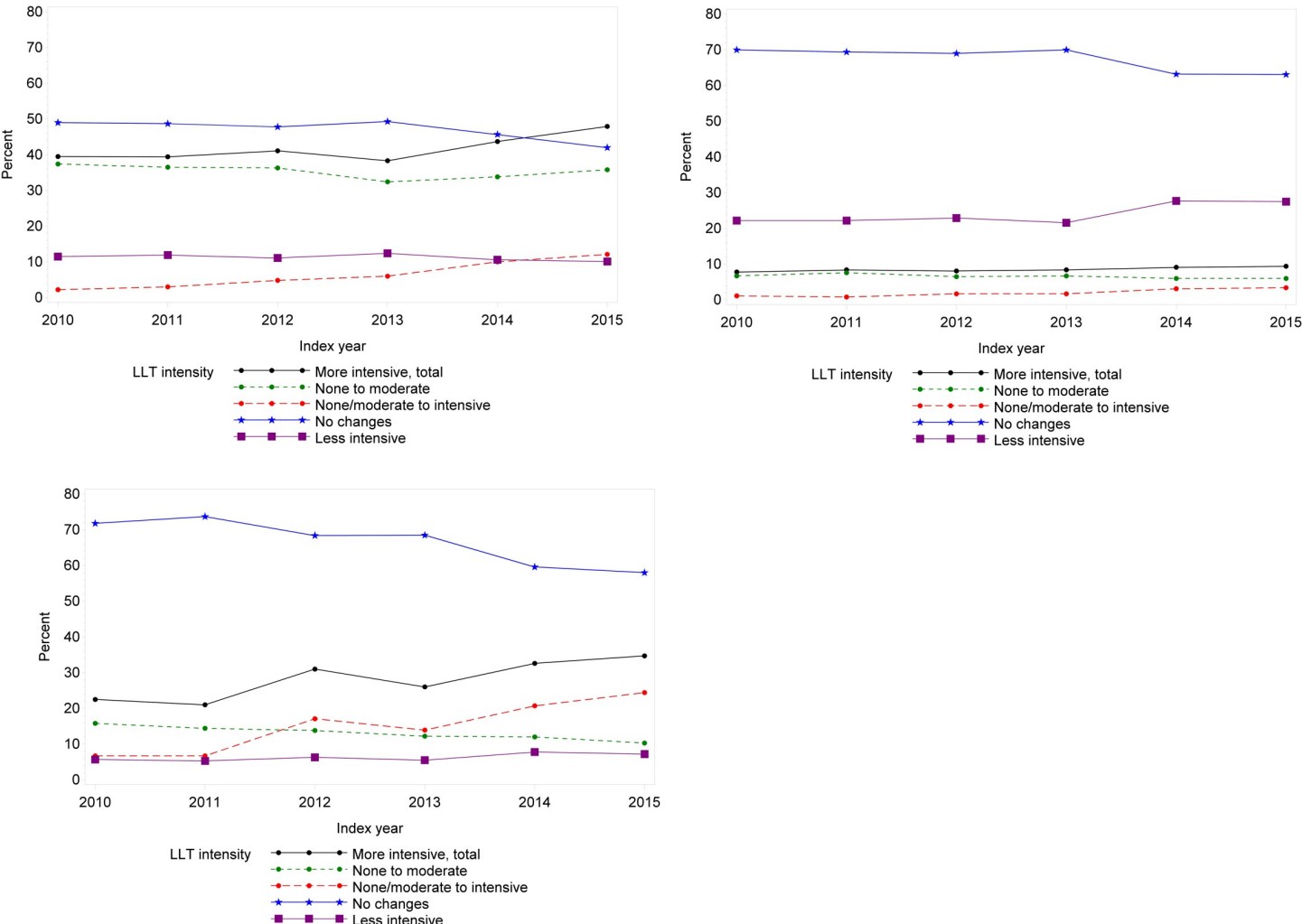

**Fig 4.** Development of progressively more intensive LLT treatment in the ASCVD population during the study period (1 January 2010–31 December 2015) given as proportion of individuals (A) 180 days before admission with incident ASCVD to 90 days after discharge, (B) 91–180 days and (C) 181–365 days after discharge, N = 11,997 individuals with an incident ASCVD.

revised guidelines with treatment goal at 1.4 mmol/L, less than 10% achieved the goal within 180 days, and less than 30% within 730 days among individuals with index date in 2015.

An increase in LLT initiation during the first year after ASCVD hospitalization was seen. The largest increase was seen in initiation of intensive LLT during the study period. However, only a minor increase was seen in initiation of any LLT in the study period. No improvement in LLT initiation was seen in the period before index date. Characteristics associated with a high odds for initiation of LLT (moderate or intensive) from admission to 90 days after discharge included type of ASCVD event (AMI and IS), younger age, no comorbidities, and no LDL-C measurements before index date. In the study period, LLT initiation increased to 75% among individuals with AMI and IS, while only 25% of individuals with other ASCVD diagnoses in 2015 initiated LLT.

Some improvement was also seen in the development towards more intensive LLT within the first year after ASCVD hospitalization. An increase in individuals changing to more

intensive LLT was seen in the period from before an ASCVD event to 90 days after the event, and in the period 181 to 365 day after an ASCVD event.

## Comparison with other studies

Our findings are in line with previous studies showing an increased proportion of individuals with LDL-C goal achievement and LLT initiation in populations at very high risk of CV events [23–27].

Rodriguez *et al.* examined trends in LDL-C goal achievement and use of high-potency statins in an American population with an ASCVD diagnosis and at least one LDL-C measurement in 2004–2012 [26]. The monthly percentage of individuals achieving LDL-C<1.8 mmol/L (70 mg/dL) increased from 11.1% to 27.3% and the percentage of individuals with LLT increased from 61.4% to 70.5%. We found similarly an increased LDL-C goal achievement within the first 6 months and similar treatments patterns.

Lamprecht *et al.* examined trends in lipid management in a population of individuals in Colorado, USA with acute or chronic ASCVD in the period 2007–2016 [24]. The proportion of individuals achieving LDL-C goal <1.8 mmol/L (70 mg/dL) increased from 39% to 54%. This is similar to the findings in the present study. The proportion of individuals receiving LLT was stable at approximately 87% in the study in Colorado, which is a larger proportion than seen in the present study.

Fox *et al.* examined LDL-C goal achievement in Germans with ASCVD and at least one LDL-C measurement and in statin (moderate or intensive) therapy at inclusion in 2012–2014 [27]. The proportion of individuals achieving LDL-C goal <1.8 mmol/L (70 mg/dL) during 1-year follow-up was 19.5%. Only a limited number of individuals changed LLT during the study period. In the present study, we found a 1-year LDL-C goal achievement increasing from 18.6%to 31.8%.

Tattersall *et al.* and Wongsalap *et al.* also studied trends in LDL-C goal achievement [23, 25]. However, these studies focused on a high-risk population in 1999–2008 and a population of individuals with acute coronary disease in 2013–2017, respectively. These studies also found some improvements during the study periods.

## Strengths

This population-based cohort study has several strengths including a large, representative sample of individuals with incident ASCVD at very high risk of a CV event. This limits the impact of selection bias [35], increased the precision of the parameter estimates, and ensures a high level of completeness and validity [29, 30, 33]. The data sources used in the present study (i.e., laboratory database, prescription, and patient registers) are reliable and validated with a high degree of completeness and validity [29, 30, 32]. This strengthened the validity of the study and the findings. Furthermore, we followed a cohort of individuals at individual-level up to two years after index date which enabled us to establish temporal relationships between ASCVD hospitalization, LLT initiation and LDL-C goal achievement. This is preferable compared to previous studies using repeated cross-sectional studies [23, 25]. Finally, appropriate statistical analyses were applied which also strengthened the validity of the study and the findings.

## Limitations

The study also has some limitations. The health registers do not contain data that might explain the patterns identified. Thus, we had no information on reasons for a suboptimal LLT. We had no information of health behavior (such as diet and physical activity) which may

influence LLT initiation and LDL-C goal achievement. However, we included information about income in the second supplementary analysis of sociodemographic and clinical characteristics associated with LLT initiation. Income (as a measure of socioeconomic status) and health behavior are to some extent correlated. The available prescription data regarding LLT are based on prescription redemption but without data on actual issuing a prescription. Data include information on date and amount of drug prescribed, but do not inform on the specific dosage to be used by the individual user. Neither do we have information about the actual consumption of drugs. Furthermore, 10% of the individuals with an incident ASCVD were excluded due to the lack of LDL-C measurements either before or during ASCVD hospitalization. In addition to individuals excluded due to the different exclusion criteria, we excluded individuals with a missing or invalid personal identification number. However, it concerned a limited number of individuals (N = 59) and is considered not to have any impact on the results. Another limitation is inclusion of lipid measurements only from a single laboratory in Denmark. Including all laboratories in Denmark would have strengthened the study and increased the generalizability of the findings. However, utilizing data from a single laboratory could also be an advantage regarding homogeneity and comparability. Finally, by identifying individuals with LLT initiation, we only used information of first prescription redemption, and longitudinal medication adherence was not tracked.

## Implications

Our findings have clinical and public health implications. The ESC/EAS guidelines [8] consider individuals with the ASCVD diagnoses included in the present study to be at a very high CV risk. Furthermore, the guidelines recommend individuals at very high risk of a CV event to initiate LLT to reduce LDL-C (<1.4 mmol/L according to the 2019 guidelines). The findings in the present study indicate a need to focus on clinical practice of individuals with ASCVD to limit the risk of further CV events and mortality. A much larger proportion of individuals with UA, SA, PAD, or coronary revascularization (e.g., PCI and CABG) should initiate LLT. A focus is also needed on individuals at older age and individuals with comorbidities (CKD, DM) where only a low proportion of individuals initiates LLT.

It is recommended to replicate the study with other and larger populations to examine the generalizability of the findings. Furthermore, it is recommended to repeat the study to monitor the development and evaluate improvement in LLT initiation and LDL-C goal achievement in the ASCVD population in order to limit the risk of a CV event.

## Conclusion

In a large population-based cohort study of LDL-C goal achievement and LLT initiation in individuals with incident ASCVD, the proportion achieving LDL-C<1.8 mmol/L (70 mg/dL) within the first 2 years after index date increased from 40.5% in 2010 to 50.6% in 2015. With the revised treatment goal of 1.4 mmol/L, less than 10% of the individuals with ASCVD achieved the goal with 180 days, and less than 25% within 730 days after index date. The proportion of individuals initiating LLT within the year after index date increased, especially for intensive LLT initiation, although only one third were initiating intensive LLT in 2015. Among individuals with AMI or IS, 75% initiated LLT with 90 days after index, whereas individuals with other ASCVD diagnoses or coronary revascularization only 25% initiated LLT within 90 days. Although trends show an improvement in LDL-C goal achievement, 49.4% of the individuals at very high risk of a CV event did not achieve the LDL-C goal within 2 years after ASCVD hospitalization.

## Supporting information

**S1 File.**
(PDF)

## Acknowledgments

We would like to acknowledge Jan Helldén from the Department of Clinical Biochemistry and Pharmacology, Odense University Hospital, for his contribution to extraction of laboratory data.

**Ethics approval and consent to participate**

The study was approved by the Danish Data Protection Agency (record number 2015-41-4130) and the Danish Health and Medicines Authority.

According to Danish law, register-based studies can be performed without consent from the subjects if the data processing takes place with the only purpose of performing statistical or scientific studies of significant public health concerns and where the processing is required to perform these studies. Before data collection, data management, and data analyses, approval was obtained from the relevant national data agencies required. The Act on Processing of Personal Data (Act No. 429 of 31 May 2000 with amendments) is the legal foundation for analyses of register-based data in Denmark [36].

## Author Contributions

**Conceptualization:** Annette Kjær Ersbøll, Marie Skov Kristensen, Mads Nybo, Kristian Handberg Mikkelsen, Gunnar Gislason, Mogens Lytken Larsen, Anders Green.

**Data curation:** Annette Kjær Ersbøll, Marie Skov Kristensen.

**Formal analysis:** Annette Kjær Ersbøll.

**Project administration:** Annette Kjær Ersbøll.

**Validation:** Annette Kjær Ersbøll, Mads Nybo, Anders Green.

**Writing – original draft:** Annette Kjær Ersbøll.

**Writing – review & editing:** Annette Kjær Ersbøll, Marie Skov Kristensen, Mads Nybo, Simone Møller Hede, Kristian Handberg Mikkelsen, Gunnar Gislason, Mogens Lytken Larsen, Anders Green.

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
