## [Decision Letter · Decision Letter 0]

22 Feb 2022

PONE-D-21-35228Trends in low-density lipoprotein cholesterol goal achievement and changes in lipid-lowering therapy after incident atherosclerotic cardiovascular disease: Danish cohort study.PLOS ONE

Dear Dr. Ersbøll,

Thank you for submitting your manuscript to PLOS ONE. After careful consideration, we feel that it has merit but does not fully meet PLOS ONE’s publication criteria as it currently stands. Therefore, we invite you to submit a revised version of the manuscript that addresses the points raised during the review process.

We look forward to receiving your revised manuscript.

Kind regards,

Arturo Cesaro, MD

Academic Editor

PLOS ONE

Journal Requirements:

“This work was supported by Sanofi Aventis Denmark A/S; Applied Economics and Health Research (ApHER).

This paper is based on data originating from a study conducted for Applied Economics and Health Research (ApHER) as an independent research institute and funded by Sanofi Aventis Denmark A/S.”

We note that you have provided additional information within the Funding Section that is not currently declared in your Funding Statement. Please note that funding information should not appear in other areas of your manuscript. We will only publish funding information present in the Funding Statement section of the online submission form.

“This work was supported by Sanofi Aventis Denmark A/S and Applied Economics and

Health Research (ApHER).

This paper is based on data originating from a study conducted for Applied Economics and Health Research (ApHER) as an independent research institute and funded by Sanofi Aventis Denmark A/S.

www.sanofi.dk

http://appliedeconomics.dk/

The sponsors (Sanofi) was involved in the Conceptualization and the final review and editing of the manuscript”

“I have read the journal's policy and the authors of this manuscript have the following competing interests:

When this study was performed, KH Mikkelsen was an employee of Sanofi Aventis Denmark A/S

A Green and SM Hede were associated with ApHER that received funding from Sanofi Aventis Denmark A/S during the conduct of the study, and they declare no other relationships or activities outside the submitted work

GG declare that he has received funding from Bristol Myers Squibb, Bayer, Boehringer Ingelheim and Pfizer outside the submitted work.

All other authors declare no conflict of interest.”

Additional Editor Comments:

You are invited to consider the reviewers' comments, reported at the end of this letter, and to revise your manuscript accordingly. In the letter accompanying your resubmission, please explain your response to each of the comments. Please observe the word count and citation style. For further details, please consult the Instructions for Authors on the website.

Reviewers' comments:

Reviewer's Responses to Questions

**Comments to the Author**

1. Is the manuscript technically sound, and do the data support the conclusions?

Reviewer #1: Yes

Reviewer #2: Yes

2. Has the statistical analysis been performed appropriately and rigorously? 

Reviewer #1: Yes

Reviewer #2: Yes

3. Have the authors made all data underlying the findings in their manuscript fully available?

Reviewer #1: Yes

Reviewer #2: Yes

4. Is the manuscript presented in an intelligible fashion and written in standard English?

Reviewer #1: Yes

Reviewer #2: Yes

5. Review Comments to the Author

Reviewer #1: The authors presented data to investigate trends in LDL-C goal achievement, initiation of lipid-lowering therapy and changes in LLT intensity in a population with atherosclerotic cardiovascular disease. The topic is of interest.

The authors found that an increased proportion achieved LDL-C goal within the first 2 years after index date.

General comments:

1) While the paper is within the word limit, I feel that readability would be enhanced by being more succinct and shortening the word count.

Detailed comments:

2) Please describe the inclusion and exclusion criteria more in detail.

3) The authors consider LDL-C assays up to 18 months before the event. They do not believe this is a very large time frame for an assessment of the lipid profile? Please specify.

4) What definition of PAD did they use?

5) Dosing timepoints were different for different patients. This is an important limitation. Don't the authors think they should mention it?

6) The authors mention "other non-statins" several times. What therapies are they referring to? What is meant by "combination therapies"? Isn't ezetimibe already in combination with statins? Please specify this point.

7) The results seem to run counter to international data showing reduced adherence with respect to statin intake manifested by poor target attainment. How do the authors explain this difference?

8) There are errors in the calculation of some percentages in Table 1. Please recheck the section "Lipid lowering therapy within 180 days before ASCVD hospitalalization" and the " Intensity of lipid lowering therapy within 180 days before ASCVD hospitalization " section.

9) The authors should at least mention new therapeutic strategies that allow the ambitious new targets to be reached. Reference is made to PCSK9 inhibitors and their effects on adherence and quality of life.

(doi: 10.1056/NEJMoa1801174; 10.1056/NEJMoa1615664; 10.1016/j.phrs.2019.03.021; 10.1177/2047487319839179; 10.2459/JCM.0000000000000611)

Reviewer #2: The study analyses the proportions of high cardiovascular risk patients achieving the recommended therapeutic target for LDL-C after starting lipid-lowering therapy (LLT) following an acute cardiovascular event in a large population sample from Funen Island in Denmark.

The manuscript is overall well written and the statistical analysis is properly conducted. In addition, the study includes a very large population sample, which render the main findings of the analysis of potential clinical relevance.

Some methodological aspects, however, remain unclear to me and should be better defined. I have some comments and suggestions to propose:

#1. It would be very useful for the readers to have also the converted units (mg/dl), eventually between parentheses, for each value reported within the text. This would render the manuscript more easy to follow for those who are less confident with international units.

#2. In the abstract, instead of reporting the proportions of patients who started LLT or intensive LLT or the proportions of patients who changed LLT intensity, it would be better to have the answers to the following questions: 1) which was the proportion of patients achieving the predefined LDL-C targets for those who started LLT within 90 -180 days?; 2) which was the proportion of patients achieving the LDL-C targets for those who started intensive LLT?; 3) how many patients who changed LLT intensity had their LLT up-titrated and how many had their LLT down-titrated? There were any difference between these two groups in terms of achievement of the predefined LDL-C targets?

#3. Some recent studies can be added to support the statement reported on page 5, lines 86-88, that a large proportions of high risk individuals do not achieve the recommended therapeutic targets for LDL-C and other lipid parameters in the real world practice. For example, the following references should be added: Cardiovasc Diabetol. 2021 Jul 16;20(1):144. doi: 10.1186/s12933-021-01338-y. + Atherosclerosis 2019 Jun;285:40-48. doi: 10.1016/j.atherosclerosis.2019.03.017 + Clin Cardiol. 2021 Nov;44(11):1575-1585. doi: 10.1002/clc.23725.

#4. Which were the main reasons for not starting LLT (or intensive LLT)? On page 10, lines 206-207, it is stated that proportion of individuals who initiated LLT within the first 90 days after index (event) increased from 48.6 to 56.0%; in other words, about half of the study population did not receive any LLT therapy after acute cardiovascular events. Which were the main reasons for these missed prescriptions in very high risk individuals?

#5. In line with the previous comment, it is not clear to me why not all patients with acute cardiovascular event did not have any LLT prescribed at hospital discharge and why they should start LLT after 90-180 days or later on. Please clarify which were the clinical indications, since they are not in line with recommendations from current international guidelines.

#6. In line with comment #2, in the conclusive remarks more emphasis should be devoted to the proportions of patients achieving the recommended therapeutic targets after hospitalization for acute cardiovascular events, rather than on absolute proportions of patients starting LLT or changing LLT intensity. Given the fact that very low proportions of patients at high cardiovascular risk were treated for high cholesterol levels before the index event, it would be easy to expect a trend toward increase in LLT prescriptions after hospitalizations, mostly in the recent years.

6. PLOS authors have the option to publish the peer review history of their article (what does this mean?). If published, this will include your full peer review and any attached files.

Reviewer #1: No

Reviewer #2: No

---

## [Author Response · Author response to Decision Letter 0]

21 Jun 2022

Thank you for comments and suggestions for revision of the manuscript.

We have addresses all comments and revised the manuscript accordingly. I point-to-point revision note is uploaded with the revised manuscript with a detailed description of the changes we have made in the revised manuscript and explanations and responses to the reviewers

---

## [Decision Letter · Decision Letter 1]

4 Nov 2022

PONE-D-21-35228R1Trends in low-density lipoprotein cholesterol goal achievement and changes in lipid-lowering therapy after incident atherosclerotic cardiovascular disease: Danish cohort study.PLOS ONE

Dear Dr. Ersbøll,

Thank you for submitting your manuscript to PLOS ONE. After careful consideration, we feel that it has merit but does not fully meet PLOS ONE’s publication criteria as it currently stands. Therefore, we invite you to submit a revised version of the manuscript that addresses the points raised during the review process.

 Thank you for resubmitting your work. We are interested in reviewing a revised version of your manuscript after addressing the comments and concerns from the reviewers below. 

We look forward to receiving your revised manuscript.

Kind regards,

Fares Alahdab

Academic Editor

PLOS ONE

Reviewers' comments:

Reviewer's Responses to Questions

**Comments to the Author**

1. If the authors have adequately addressed your comments raised in a previous round of review and you feel that this manuscript is now acceptable for publication, you may indicate that here to bypass the “Comments to the Author” section, enter your conflict of interest statement in the “Confidential to Editor” section, and submit your "Accept" recommendation.

Reviewer #2: (No Response)

Reviewer #3: (No Response)

Reviewer #4: All comments have been addressed

Reviewer #5: (No Response)

Reviewer #6: All comments have been addressed

Reviewer #7: All comments have been addressed

Reviewer #8: All comments have been addressed

2. Is the manuscript technically sound, and do the data support the conclusions?

Reviewer #2: Yes

Reviewer #3: Partly

Reviewer #4: (No Response)

Reviewer #5: Yes

Reviewer #6: Yes

Reviewer #7: Yes

Reviewer #8: Yes

3. Has the statistical analysis been performed appropriately and rigorously? 

Reviewer #2: Yes

Reviewer #3: No

Reviewer #4: (No Response)

Reviewer #5: Yes

Reviewer #6: Yes

Reviewer #7: Yes

Reviewer #8: Yes

4. Have the authors made all data underlying the findings in their manuscript fully available?

Reviewer #2: Yes

Reviewer #3: (No Response)

Reviewer #4: (No Response)

Reviewer #5: Yes

Reviewer #6: No

Reviewer #7: Yes

Reviewer #8: No

5. Is the manuscript presented in an intelligible fashion and written in standard English?

Reviewer #2: Yes

Reviewer #3: Yes

Reviewer #4: (No Response)

Reviewer #5: Yes

Reviewer #6: Yes

Reviewer #7: Yes

Reviewer #8: Yes

6. Review Comments to the Author

Reviewer #2: Authors addressed some of the comments proposed and modified the manuscript accordingly. Please consider specific responses to the remaining issues:

#3. Some recent studies can be added to support the statement reported on page 5, lines 86-88, that a large proportions of high risk individuals do not achieve the recommended therapeutic targets for LDL-C and other lipid parameters in the real world practice. For example, the following references should be added: Cardiovasc Diabetol. 2021 Jul 16;20(1):144. doi: 10.1186/s12933-021-01338-y. + Atherosclerosis 2019 Jun;285:40-48. doi: 10.1016/j.atherosclerosis.2019.03.017 + Clin Cardiol. 2021 Nov;44(11):1575-1585. doi: 10.1002/clc.23725.

#4. Which were the main reasons for not starting LLT (or intensive LLT)? On page 10, lines 206-207, it is stated that proportion of individuals who initiated LLT within the first 90 days after index (event) increased from 48.6 to 56.0%; in other words, about half of the study population did not receive any LLT therapy after acute cardiovascular events. Which were the main reasons for these missed prescriptions in very high risk individuals?

#5. In line with the previous comment, it is not clear to me why not all patients with acute cardiovascular event did not have any LLT prescribed at hospital discharge and why they should start LLT after 90-180 days or later on. Please clarify which were the clinical indications, since they are not in line with recommendations from current international guidelines.

Reviewer #3: While the revised paper showed the modest changes of lipid treatment during a period time in the past, three are still some issues remained to be clarified including those suggested in the previous review.

1. Since only one population is evaluated, it is not known if this is a usual condition or region-specific condition. It is well known that many factors including the budget administration, medicare system, national/regional treatment guidelines, the specialty of the doctors for patient care, and so on may significantly impact on the real world lipid treatment. The above information and data analysis may be required to improve the background, the understanding and the potential implication of the current findings.

2. It seems all types of the patients including acute coronary syndrome, stable symptomatic CAD, and silent CAD were enrolled according to the definition of incident atherosclerotic cardiovascular disease in the current study. However, it is well known that the indication for lipid-lowering treatment especially the use of stratin could be quite different in case of acute coronary syndrome vs. stable CAD in clinical practice according to the contemporary guideline. It is critical to divide the patients according to acute coronary syndrome, stable symptomatic CAD, and silent CAD by their clinical presentation to see if there is difference in lipid lowering treatment as indicated. Hopefully such analysis may answer the questions of the reviewers and improve the understandings on the current data.

3. As mentioned by the authors, there are significant portion of the patients still without (adequate) lipid lowering treatment in the current observations. It is very important to the potential implication of the current findings. Some additional analysis may be required to see if there is difference in the patient characteristics (age, gender, and some on), disease patterns, living areas, medical resources, economic status, the specialty of the in charge doctors, and so on between patients with and those without (adequate) lipid lowering treatment. It should be helpful to improve the future lipid control in such patients.

4. Further discussion to explain and interpret at the current findings is indicated, which may be possible if the above-mentioned data and statistical analysis are available.

Reviewer #4: The authors have considered the reviewers’ suggestion and improved the paper accordingly. I’ve no further comments on it.

Reviewer #5: My major concern is that the 2019 revised guidelines recommend an even lower treatment goal (as LDL-C< 54 mg/dL) for individuals at very high risk. Thus, the results of the study may appear obsolete from this point of view. Obviously, they describe a trend that is independent from target values, but it could be very interesting to have some data on the initial and final proportion of individuals reaching this more ambitious target in this population, at least in form of supplementary table. Besides, this should be indicated as a major limitation of the study.

A procedure of coronary of angiography (CAG) is not an equivalent of ASCVD. Namely, no coronary plaques could be demonstrated. So, the inclusion of these subjects should be also indicated as a study limitation.

Minor concerns:

According to the most recent guidelines, your population should be considered at very high risk and not at high risk. This should be specified.

Definition of CKD should be specified.

In table 1, the indicated percent for total single ASCVD is relative to type of ASCVD, whereas that during or before hospitalization is relative to the proportion between these periods. This may be confounding and may be specified with a note or in the table legend. The same is valid for the other variables (calendar year, age group, and so on).

I think that the note on lipid measurements within 18 months before ASCVD hospitalization is e and not a; besides, notes b, c and e are indicated in the legend but not in the main text of the table.

In the methods, you affirm to exclude patients with a previous event in the period 197-2009; this should be clearly specified in the study flow-chart.

Discussion is clear but very synthetic; I think it may be enlarged focusing on the many study results.

Reviewer #6: This paper reports data obtained in the years 2010 to 2015. In these years, the target for LDL-C in high-risk patients, who are included into this study, was set at 1.8 mmol/L. As the authors indicate in their introduction, this target has been reduced to 1.4 mmol/L in the 2019 guideline. In the latter guidelines, an even lower target was recommended for patients who suffered from several ASCVD events within two years. This aspect has not been taken into consideration in this manuscript. Authors looked at first-ever ASCVD cases only. Lipoprotein(a) was not mentioned at all.

Data of this manuscript essentially only describe the treatment with statins and ezetimibe. The reviewer did not see numbers for the use of fibrates, bile acid sequestrants or nicotinic acid. LLT was categorized by intensity groups which have been defined by the number of prescription redemptions. Individuals were followed for a maximum of two years with censoring at the end of the study period.

The major finding of this study is that the proportion of individuals who achieved LDL-C<1.8 mmol/L (70 mg/dL) during the first 730 days after ASCVD hospitalization increased from 40.5% in 2010 to 50.6% in 2015. But unfortunately, as discussed by the authors in their responses to the comments of the previous two reviewers, no reason for this increase can be described. This is also discussed in the chapter on limitations of the study.

The scientific significance of the data given for different time periods (e g 181 – 365 days) is rather minimal.

It is astonishing to read the 10 % of the individuals with an incident ASCVD were excluded due to the lack of LDL-C measurements either before or during hospitalization.

The figures are difficult to read – color is needed to differentiate the curves.

Reviewer #7: The authors correctly responded to the reviewers' comments and corrected the errors. The manuscript has improved readability and the message is now sharper in focus and clearer.

Reviewer #8: Appreciate all of your corrections. The paper reads well. The big question ,of course, is why weren’t a higher proportion of patients able to reach a guideline recommended goal of LDL-C <1.8 mmol/L (70 mg/dl) within 2 years of a diagnosis of incident ASCVD. Hope that’s a source of further inquiry

7. PLOS authors have the option to publish the peer review history of their article (what does this mean?). If published, this will include your full peer review and any attached files.

Reviewer #2: No

Reviewer #3: No

Reviewer #4: **Yes: **Arrigo F.G Cicero

Reviewer #5: No

Reviewer #6: No

Reviewer #7: **Yes: **Lluís Masana

Reviewer #8: No

---

## [Author Response · Author response to Decision Letter 1]

3 Apr 2023

Dear Editor and Reviewers

Thank you very much for reviewing the manuscript, your comments and suggestions. We have addressed all comments and suggestions and revised the manuscript accordingly. We have described all changes in the document: "Revision note" included in the submission.

---

## [Decision Letter · Decision Letter 2]

17 Apr 2023

PONE-D-21-35228R2Trends in low-density lipoprotein cholesterol goal achievement and changes in lipid-lowering therapy after incident atherosclerotic cardiovascular disease: Danish cohort study.PLOS ONE

Dear Dr. Ersbøll,

Thank you for submitting your manuscript to PLOS ONE. After careful consideration, we feel that it has merit but does not fully meet PLOS ONE’s publication criteria as it currently stands. Therefore, we invite you to submit a revised version of the manuscript that addresses the points raised during the review process. Thank you for submitting your paper to our journal. We appreciate the effort and time you have invested in your research. After a thorough review, we believe your study has merit and potential to contribute to the field. We would like to move forward toward acceptance after one (hopefully last) round of minor edits. Below you will find a few comments raised regarding the strengths and weaknesses of the paper. We understand many of the weaknesses raised require substantial work to improve this manuscript, which makes it more suitable for future work. But we would like that you make sure those weaknesses are addressed in the current manuscript (if possible), and expressed explicitly in the Limitations section (if unable to remedy).

In order to help you improve the quality of your manuscript, we have provided detailed feedback and suggestions. We kindly ask you to consider these comments and critiques carefully as you revise your paper. Addressing these points will not only strengthen the methodology and discussion but also enhance the overall clarity, organization, and significance of your study.

Once you have made the necessary revisions, please resubmit your manuscript for further review. We look forward to receiving your revised paper and evaluating its potential for publication in our journal.

Thank you for considering our feedback, and we hope to receive your revised manuscript soon.

We look forward to receiving your revised manuscript.

Kind regards,

Fares Alahdab, MD, MSc

Academic Editor

PLOS ONE

Journal Requirements:

Reviewers' comments:

Reviewer's Responses to Questions

**Comments to the Author**

1. If the authors have adequately addressed your comments raised in a previous round of review and you feel that this manuscript is now acceptable for publication, you may indicate that here to bypass the “Comments to the Author” section, enter your conflict of interest statement in the “Confidential to Editor” section, and submit your "Accept" recommendation.

Reviewer #1: All comments have been addressed

Reviewer #2: (No Response)

2. Is the manuscript technically sound, and do the data support the conclusions?

Reviewer #1: (No Response)

Reviewer #2: Partly

3. Has the statistical analysis been performed appropriately and rigorously? 

Reviewer #1: (No Response)

Reviewer #2: Yes

4. Have the authors made all data underlying the findings in their manuscript fully available?

Reviewer #1: (No Response)

Reviewer #2: No

5. Is the manuscript presented in an intelligible fashion and written in standard English?

Reviewer #1: (No Response)

Reviewer #2: Yes

6. Review Comments to the Author

Reviewer #1: The authors have satisfactorily responded to the problems which I raised in my previous review.

The supplementary analyses clearly improved the quality of the manuscript.

The nature of this study based on prescription redemptions limits possible interpretations with respect to reasons for this low rate of achievement of target values for LDL-C. Physicians attitude towards the initiation of LLT cannot be evaluated in this way, but it may have had a major impact on the performance of LLT in a given patient.

Data have been gathered about 10 years ago – since that time new developments of drugs were made, ASCVD is now more in the focus of the medical community.

Reviewer #2: This is a study that examined trends in lipid-lowering therapy (LLT) and low-density lipoprotein cholesterol (LDL-C) goal achievement among individuals with incident atherosclerotic cardiovascular disease (ASCVD) in Funen Island, Denmark, between 2010 and 2015. The study found an overall improvement in LDL-C goal achievement and an increased initiation of LLT during the first year after ASCVD hospitalization, particularly in the initiation of intensive LLT. However, a large proportion of individuals did not achieve the LDL-C goal within 730 days after index date, and less than 30% of individuals achieved the revised LDL-C goal of 1.4 mmol/L within that same timeframe. The study also identified characteristics associated with higher odds for LLT initiation, including type of ASCVD event, younger age, no comorbidities, and no LDL-C measurements before index date. The study's findings were generally consistent with previous studies that examined trends in LDL-C goal achievement and LLT initiation in populations at high risk of cardiovascular events.

Study Design: The study utilized a retrospective cohort design which is appropriate for investigating the relationship between exposure and outcome in large populations. The study design enabled the researchers to establish temporal relationships between LDL-C goal achievement and LLT initiation, and ASCVD hospitalization.

Data Sources: The study utilized several sources of data including the Danish National Patient Register, the Danish National Prescription Registry, and the Danish Civil Registration System. These are reliable and validated sources of data, and their use strengthens the validity of the study.

Study Population: The study population was individuals with incident ASCVD on Funen Island, Denmark. The sample size was large (n=11,997), and the sample was representative of the general Danish population. However, the study may not be generalizable to populations outside Denmark.

Exposure and Outcome Measures: The study utilized LDL-C measurements as the primary exposure and LLT prescription redemption as a secondary exposure. The outcome measures were LDL-C goal achievement and LLT initiation. The use of these measures is consistent with current guidelines and strengthens the validity of the study.

Statistical Analysis: The statistical analysis was rigorous and appropriate. The researchers utilized multivariable logistic regression models to determine the odds of achieving LDL-C goal and initiating LLT. They also utilized Kaplan-Meier curves to determine the time to LDL-C goal achievement and LLT initiation. These methods were appropriate and strengthened the validity of the study.

Strengths: The authors describe a few legitimate strengths in their paper, here are a few more.

1. Large sample size: The study had a large sample size, which increased the precision of the estimates.

2. Use of reliable data sources: The study utilized reliable and validated data sources which strengthened the validity of the study.

3. Appropriate statistical analysis: The statistical analysis was rigorous and appropriate, which strengthened the validity of the study.

4. Temporal relationships: The study design enabled the researchers to establish temporal relationships between LDL-C goal achievement, LLT initiation, and ASCVD hospitalization.

Weaknesses: The authors describe a few legitimate limitations in their paper, here are a few more.

1. Limited generalizability: The study was conducted on Funen Island, Denmark, which may limit the generalizability of the findings to other populations.

2. Exclusion criteria: The study excluded individuals with missing data which may have resulted in a selection bias. But there is no better alternative that the authors could easily do at this point, so this was the appropriate approach to follow.

3. Limited information on lifestyle factors: The study did not include information on lifestyle factors such as diet and exercise which may have influenced LDL-C goal achievement and LLT initiation.

4. No adjustment for confounding by indication: The study did not adjust for confounding by indication, which may have influenced the estimates.

Ways for the authors to improve their paper: (mostly in future work)

1. The authors could include information on lifestyle factors such as diet and exercise to determine their influence on LDL-C goal achievement and LLT initiation.

2. The authors could adjust for confounding by indication to increase the validity of the estimates.

3. The authors could conduct the study on a larger and more diverse population to increase the generalizability of the findings.

4. The authors could consider using other measures of exposure such as total cholesterol or HDL-C to determine their influence on LDL-C goal achievement and LLT initiation.

7. PLOS authors have the option to publish the peer review history of their article (what does this mean?). If published, this will include your full peer review and any attached files.

Reviewer #1: **Yes: **Arrigo Francesco Giuseppe Cicero

Reviewer #2: No

---

## [Author Response · Author response to Decision Letter 2]

14 May 2023

A revision note has been uploaded with a point-to-point response to all comments from the reviewers including a description of changes introduced in the manuscript and supplementary material.

---

## [Editor Report · Decision Letter 3]

16 May 2023

Trends in low-density lipoprotein cholesterol goal achievement and changes in lipid-lowering therapy after incident atherosclerotic cardiovascular disease: Danish cohort study.

PONE-D-21-35228R3

Dear Dr. Ersbøll,

We’re pleased to inform you that your manuscript has been judged scientifically suitable for publication and will be formally accepted for publication once it meets all outstanding technical requirements.

Kind regards,

Fares Alahdab, MD, MSc

Academic Editor

PLOS ONE
---

## [Editor Report · Acceptance letter]

18 May 2023

PONE-D-21-35228R3 

Trends in low-density lipoprotein cholesterol goal achievement and changes in lipid-lowering therapy after incident atherosclerotic cardiovascular disease: Danish cohort study. 

Dear Dr. Ersbøll:

I'm pleased to inform you that your manuscript has been deemed suitable for publication in PLOS ONE. Congratulations! Your manuscript is now with our production department. 

Kind regards, 

on behalf of

Dr. Fares Alahdab 

Academic Editor

PLOS ONE